# Methods for Studying Endocytotic Pathways of Herpesvirus Encoded G Protein-Coupled Receptors

**DOI:** 10.3390/molecules25235710

**Published:** 2020-12-03

**Authors:** Maša Mavri, Katja Spiess, Mette Marie Rosenkilde, Catrin Sian Rutland, Milka Vrecl, Valentina Kubale

**Affiliations:** 1Department of Anatomy, Histology with Embryology and Cytology, Institute of Preclinical Sciences, Veterinary Faculty, University of Ljubljana, Gerbičeva 60, 1000 Ljubljana, Slovenia; masa.mavri@vf.uni-lj.si (M.M.); milka.vrecl@vf.uni-lj.si (M.V.); 2Department of Biomedical Sciences, Faculty of Health and Medical Sciences, University of Copenhagen, 2200 Copenhagen, Denmark; kspiess@sund.ku.dk (K.S.); rosenkilde@sund.ku.dk (M.M.R.); 3School of Veterinary Medicine and Science, Medical Faculty, Sutton, Bonington Campus, University of Nottingham, Sutton Bonington LE12 5RD, UK; Catrin.Rutland@nottingham.ac.uk

**Keywords:** endocytosis, G-protein coupled receptors, herpesvirus, methods

## Abstract

Endocytosis is a fundamental process involved in trafficking of various extracellular and transmembrane molecules from the cell surface to its interior. This enables cells to communicate and respond to external environments, maintain cellular homeostasis, and transduce signals. G protein-coupled receptors (GPCRs) constitute a family of receptors with seven transmembrane alpha-helical domains (7TM receptors) expressed at the cell surface, where they regulate physiological and pathological cellular processes. Several herpesviruses encode receptors (vGPCRs) which benefits the virus by avoiding host immune surveillance, supporting viral dissemination, and thereby establishing widespread and lifelong infection, processes where receptor signaling and/or endocytosis seem central. vGPCRs are rising as potential drug targets as exemplified by the cytomegalovirus-encoded receptor US28, where its constitutive internalization has been exploited for selective drug delivery in virus infected cells. Therefore, studying GPCR trafficking is of great importance. This review provides an overview of the current knowledge of endocytic and cell localization properties of vGPCRs and methodological approaches used for studying receptor internalization. Using such novel approaches, we show constitutive internalization of the BILF1 receptor from human and porcine γ-1 herpesviruses and present motifs from the eukaryotic linear motif (ELM) resources with importance for vGPCR endocytosis.

## 1. Importance of Endocytosis for Viral GPCRs

Endocytosis encompasses different routes by which a cell uptakes extracellular material from the surface and transports it into the cell thereby maintaining homeostasis between the extracellular and intracellular environment [1]. Nutrients, receptor-ligand complexes, extracellular matrix, cell debris, bacteria, and viruses can enter the cell by different endocytic mechanisms [2,3]. Importantly, endocytosis is also linked with cellular signaling at the plasma membrane (PM) where receptors bind specific signaling molecules and initiate internalization [4].

Receptors located at the cell surface coordinate many different physiological processes in the cell. Among them, G protein coupled receptors (GPCRs) with seven transmembrane domains (7TM) are a major family of receptors able to stimulate important intracellular signaling pathways in response to various extracellular stimuli [5]. It has been recognized that after initial activation and desensitization on the cell membrane, GPCRs subsequently enter the cell via endocytosis. Endocytosis can occur either constitutively (without ligand stimulus) or in response to certain stimuli, including growth factors, viruses (Ebola, SARS and other coronaviruses), and different ligands [6]. In the cell, further sorting of internalized GPCRs between degradation and recycling pathways occurs. Therefore, cells can tightly regulate GPCR surface availability for further signaling events [7].

Endocytosis is also a mechanism used by several herpesviruses (for example Epstein-Barr virus, Kaposi’s sarcoma associated herpesvirus, cytomegalovirus, and varicella zoster virus) for initial entry into the cell [8]. Herpesviruses are widespread DNA viruses employing a special bipartite life cycle where latent and lytic phases interchange in order to persist in the infected host for their whole life [9]. The herpesvirus virion consists of three main parts: a nucleocapsid containing linear double-stranded DNA, an envelope, and tegument. In the envelope, different glycoproteins are involved in the initial binding and endocytosis events upon infection of susceptible cells. Herpesvirus genomes encode between 100 to 200 proteins. These proteins are involved in DNA replication (e.g., DNA polymerase), viral entry, cell-to-cell spread, immunevasion, and pathogenesis. Among these regulatory proteins, herpesviruses encode GPCRs (vGPCRs). It is believed that during evolutionary processes, viruses took over genes for these receptors from their hosts and rearranged them to function in the benefit of the virus [10,11,12]. They imitate the function of endogenous human receptors and therefore use them to subvert cellular signaling, avoid cell immune responses, induce cell transformation, and support viral dissemination and replication [12,13].

Many vGPCRs resemble endogenous chemokine receptors structurally, and bind a broad spectrum of both endogenous and virally encoded chemokines, leading to activation of downstream signaling pathways [12,14]. Others are described as “orphan” vGPCRs for which no ligand has yet been identified. Additionally, BILF1 receptors, encoded by gamma-1 herpesviruses, have recently been recognized as the first immune evasive vGPCR able to downregulate surface MHC class I molecules at the cell surface [15]. Initial localization of these receptors in the cell, and additional trafficking, are important processes that control the signaling capacity of these receptors. Table 1 summarizes known viruses and viral GPCRs from the β- and γ-herpesvirus family and their known endocytic and signaling pathways.

In this article, the most common endocytic pathways employed by GPCRs to enter the cell are reviewed, alongside the novel approaches used to study GPCR mediated endocytosis and endocytic properties of most commonly studied vGPCRs: US28, ORF74, and BILF1.

## 2. Different Endocytic Pathways

In general, endocytosis is divided into two processes: pinocytosis, by which cells take in fluid and small particles, and phagocytosis, which is performed by specialised cells that have the capacity to uptake larger particles (>500 nm) such as microorganisms and cell debris (Figure 1). Pinocytosis is further divided into macro- and micropinocytosis. By macropinocytosis, cells take in extracellular fluid via large endocytic vesicles which are heterogeneously sized (200–500 nm) termed macropinosomes. In micropinocytosis, specific molecules enter the cell through smaller vesicles and can be further divided to clathrin-mediated endocytosis (70–150 nm) (CME), caveolae-mediated endocytosis (60–80 nm) or non-coated vesicles (Figure 1) [3,17].

After pinching off the PM, the majority of endocytic vesicles are fused with early endosomes where the cargo is sorted. Later, it can either recycle back to the PM or it can be directed into degradation pathways. Late endosomes are formed via fusion of early endosomes in an endosome maturation process, where the structures of membrane proteins change. After endosome maturation, the recycling to the PM stops and non-recycled material is further directed into degradation pathways [18]. Degradation takes place in lysosomes, which are formed by fusion of late endosomes and pre-existing lysosomes. During maturation, transport from the trans-Golgi network to endosomes occurs, providing newly synthesized lysosomal enzymes for the endosomes [19].

### 2.1. Clathrin-Mediated Pathway

The most extensively studied form of endocytosis is the clathrin-mediated pathway, which occurs in all mammalian cells. This pathway has traditionally been described as the most commonly used endocytic pathway for the majority of GPCRs [20]. Among vGPCRs, US28 and ORF74 use the clathrin-mediated pathway as one of the mechanisms for internalization, however, their mechanism for cell entry is promiscuous, as described later. A clathrin-coated vesicle forms on the PM and is coated with the protein clathrin on its cytosolic face. This plays a central role for receptor internalization and recycling, but also for the uptake of numerous important molecules such as nutrients, antigens, growth factors, and pathogens [21]. The CME is divided into the following stages: coat nucleation and assembly, coated pit maturation, fission, and uncoating [22].

The clathrin coat assembles on the cytosolic surface of the PM and is composed of different components: adaptor proteins, cargo, lipids, and clathrin [22]. Clathrin, the most important component of the coat, comprises of three clathrin heavy chains (CHC) and three clathrin light chains (CLC). Its recognisable structure is also termed triskelion. With clathrin polymerisation, a lattice-like structure forms around the vesicle. Since clathrin is unable to directly bind to the lipid and protein components of the PM, it requires adaptor proteins which link the clathrin coat with the PM and specific cargo assembled on PM [6,23].

Adaptor protein 2 (AP2) is the major adaptor protein which is bound to the membrane phosphatidylinositol. It consists of four subunits: two larger α and β2 and two smaller µ2 and σ2. Via the α subunit, AP2 binds to phosphatidylinositol that anchors AP2 at the PM. β2 subunit binds the heavy chain of clathrin and therefore links clathrin to the PM. The µ2 subunit recognises specific tyrosine motifs (YXXΦ; Φ represents a hydrophobic residue (V, I, L, F, W, Y, M)) on integral membrane receptors, which triggers their selective arrangement in clathrin rich areas. AP2 can bind to receptors via other adaptor proteins such as the β-arrestins which bind to the β2 subunit of AP2 [21,24]. Binding to β-arrestins is not obligatory for receptor internalization as observed in US28 and ORF74. For US28, constitutive β-arrestin independent clathrin-mediated internalization has been described in MEF (mouse embryonic fibroblasts) cells, where both β-arrestin 1 and β-arrestin 2 were depleted [25]. For ORF74, the difference in β-arrestin recruitment was observed by comparing ligand-induced and constitutive internalization. CXCL1 and CXCL8 mediated rapid endocytosis relies on recruitment of both β-arrestin-1 and -2, whereas constitutive endocytosis does not require β-arrestins, comparable to US28 [26].

Epsin and eps15 are adaptor proteins with ubiquitin-interacting motifs. Therefore, they direct ubiquitin tagged cargo into clathrin-coated vesicles. Different proteins such as AP180 and CALM mediate the size of the clathrin-coated vesicle. AP180 is found in neurons and CALM is ubiquitously spread [22].

Formation of a clathrin-coated vesicle involves PM bending, which occurs under the influence of protein endofilin, amphiphysin, and epsin. Amphiphysin binds to clathrin, AP2, and dynamin [6]. Dynamin GTPases are necessary for pinching off mature clathrin-coated vesicles. After clathrin-coated vesicles enter the cell they rapidly depolymerize their clathrin coat into units under the influence of cytosolic chaperone Hsc70, which catalyses the depolymerization [21].

Specific motifs located at the C-terminal domains of transmembrane proteins determine recruitment in clathrin-coated vesicles. There are four types of motifs recognized: YXXΦ (interacts with µ2 subunit of AP2), [DE]XXXL[LI] (recognized by β2 subunit of AP2 and α/σ2 hemicomplex), FXNPXY (interacts with µ2 subunit of AP2) and polyubiquitination (recognized by epsin and Eps15) [24,27]. In this article the internalization motifs recognized in vGPCRs US28, ORF74, and BILF1 are presented (Table 2). 

Eukaryotic linear motifs (ELMs) are protein interaction sites important for regulation of different biological roles exploited by proteins. We investigated known endocytosis-related ELMs of six vGPCR (EBV-BILF1, PLHV1-BILF1, PLHV2-BILF1, PLHV3-BILF1, HCMV-US28, and KSHV-ORF74) using ELM resource database [28]. The obtained pattern of conserved amino acids, which is a set of sequences that can be related to molecular function, is common for all and is MVIF. Probability was also calculated for each ELM class and was same for all (*p* = 0.0259) and should reflect the probability of the regular expression to be found by chance in any given protein sequence. All six receptors encode YXXΦ sorting signal, which interacts with the µ2 subunit of AP2 complex in clathrin-coated vesicles. This observation is in line with previously described observations for US28 and ORF74, where functional interaction with the AP2 complex was proven [25,29]. However, BILF1 receptors have not been functionally characterized yet in respect to their endocytic strategies. Based on these predicted ELMs observed for BILF1 receptors, it is our aim to test and describe routes these receptors use in endocytic trafficking.

Each of these vGPCRs were also checked for nucleotide base sequence similarities (BLAST, NCBI). EBV-BILF1, showed sequence similarities to other Herpesvirales (taxid: 548681), but no high homology outside of this taxid, indeed all returns showed herpesvirus 4. PLHV1-BILF1 again stayed within the Herpesvirales with 92.5% homology to PLHV2, 75.59% to PLHV3, 99.92% PLHV1 and 100% with porcine gamma-herpesvirus envelope glycoprotein B (gpB) gene, however their query covers amounted to just 81%, 50%, 3%, and 3% respectively. Hits with myotis gammaherpesvirus 8, rhinolophus gammaherpesvirus 1, and cricetid gammaherpesvirus 2 were also observed but with low query covers. PLHV2-BILF1 naturally had homologies with various porcine herpesvirus 2 sequences but also with PLHV3 (76.05% with 46% query coverage) and PLHV1 and porcine gamma-herpesvirus envelope glycoprotein B (~93% with just 4% query coverage), with bovine gammaherpesvirus 6, rhinolophus gammaherpesvirus 1, human gammaherpesvirus 8 and herpesvirus 8 type M all featuring on the distance tree. PLHV3-BILF1, showed similarities between lymphotropic herpesvirus 1 and 2 (98.3% and 78.32% with query coverage of 11% and 9% respectively). Once coverage was under 3% only molossus molossus gammaherpesvirus 1 isolate and alcelaphine gammaherpesvirus 1 strains showed similarities. HCMV-US28 naturally showed similarities with the differing human herpesvirus 5 strains, whilst KSHV-ORF74 sequence similarity matches mainly came up with human herpesvirus 8, human gammaherpesvirus Kaposi’s sarcoma-associated herpesvirus glycoprotein M (all of which were observed with 97–100% homology to differing sequences).

### 2.2. Caveolae

Caveolae are flat to flask shaped, 60–80 nm wide membrane pits, rich in the protein caveolin. They are enriched in certain cell types such as fibroblasts, smooth muscle cells, endothelial cells, and adipocytes and are involved in endocytosis and transcytosis as well as in calcium signalling and other signal transduction events. They are also involved in endocytosis of different pathogens [30]. Furthermore, internalization of different membrane components such as extracellular ligands, bacterial toxins and viruses (SV40, Polyoma viruses) can occur through caveolae [6]. Three types of caveolins (CAV 1, 2, 3) are important for caveolae formation. They all possess specific hairpin structures in both N- and C-terminus and in their long U-shaped intermembrane part. Caveolae formation with CAV 1 and 3 is closely related with lipid rafts, since these two proteins are mainly located in cholesterol, sphingolipids, and sphingomyelin rich parts of membrane. Dynamin is responsible for pinching off caveolae from the PM [30]. After entering the cell, caveolae can either fuse with early endosomes (this process depends on Rab5 protein presence), or caveosomes (this process is independent of Rab5 protein) or are recycled back onto the PM [31]. By utilizing caveolae mediated cell entry, different pathogens (bacteria, viruses) avoid degradation in lysosomes and thereby prolong their survival in cells [30].

### 2.3. Lipid Rafts

Lipid rafts are small (10–200 nm), mobile, heterogeneous, and detergent resistant domains enriched in cholesterol, sphingolipids, glycosylphosphatidylinositol (GPI-anchored protein), and glycosphingolipid. Lipid rafts play important roles in different aspects of cellular physiology, although specific mechanisms of their functions are still not clear [32]. It has been described that various immune receptors are translocated to lipid rafts upon their activation as a consequence of high signalling molecule concentration in these areas [33,34,35]. Many viruses bind to lipid rafts to initiate their further entry into the cell. They either bind to glycolipids enriched in the lipid rafts or to different viral receptors [36]. Moreover, it is also known that different bacterial toxins (Shiga toxin and Cholera toxin) and certain viruses (Polyoma virus and simian virus 40) use specific PM lipids as receptors for endocytosis [37]. It has recently been proposed that novel corona virus SARS-CoV-2 uses lipid rafts for entering into the host cell by binding ACE2 (angiotensin converting enzyme 2) which is enriched in lipid rafts, however, further studies are required [38]. It has also been suggested that viral budding of HIV can occur in lipid rafts due to their enrichment of cholesterol molecules [39].

Various lipid raft-dependent pathways have been described, with caveolae dependent endocytosis being one of them. GPI-anchored proteins, endothelin, growth factors, and glycolipids exploit lipid rafts for endocytosis [40]. A potential role of lipid rafts has also been described for US28. This receptor is enriched in detergent-resistant membrane fractions and is palmitoylated at the C-terminal end upon receptor activation. This process acts as a targeting signal that directs the receptor to caveolae. However, co-localization of US28 with caveolin was not detected indicating a raft dependent, caveolin-independent pathway for US28 [41].

## 3. Endocytic Properties of The Most Commonly Studied vGPCRs

### 3.1. Cytomegalovirus (CMV)

Cytomegalovirus is a severe virus which causes deadly infections among immunosuppressed patients [42]. Human cytomegalovirus (HCMV) open reading frames (ORF) encodes four vGPCRs: US28, US27, UL33, and UL78; the first three mimic chemokine receptor structure. Mouse and rat CMV encodes homologs of UL33 and UL78, but not US28 and US27 [12].

US28 is an extensively studied receptor that has been shown to be functionally important in various aspects of HCMV infection. It displays high constitutive (ligand independent) activity and binds a broad range of chemokines [43,44,45]. It is the first vGPCR used as an antiviral drug target to selectively kill HCMV infected cells using fusion toxin protein [46,47,48]. This strategy relies on the ability of this receptor to internalize constitutively and thereby deliver the immunotoxin intracellularly [25,47]. The majority of US28 receptors is located intracellularly in endosomes in the perinuclear region with only a small amount located in the PM (Figure 2). It was revealed that constitutive endocytosis is β-arrestin independent; nevertheless, it still employs a pathway through clathrin-coated vesicles as shown in embryonic fibroblasts acquired from β-arrestin-1 and -2 knockout mice using siRNA against the μ2 subunit of AP-2 adaptor protein [25]. When comparing the endocytosis of US28 in the control and β-arrestin knockout cells, no differences were observed [25].

It is assumed that US28 internalization could partially use caveolae or lipid rafts, since palmitoylation of US28 has been observed. Palmitoylation is a step in the receptor activation process and serves as a targeting signal for receptors to caveolae [41]. Typically, after ligand-mediated receptor activation, the C-terminal tail is phosphorylated, which eventually leads to β-arrestin binding and endocytosis. The C-terminal tail of US28 receptor contains many serine and threonine residues which represent potential phosphorylation sites. Research shows that the C-terminal end is constitutively phosphorylated, therefore it enables the constitutive activity of receptors [25].

Similar to US28, US27 and UL33 (HCMV) are located intracellularly. The intracellular localization of UL33 was however only observed in HeLa and COS-7 cells, but not in HEK-293, where the receptor was mostly located at the PM. This suggests that the receptor localization is cell type specific. Co-localization of US27-YFP/UL33-GFP with LAMP1, the marker of late endosomes/lysosomes, was shown in Hela cells, which indicates that these receptors are located in late endocytic compartments [49]. It is believed that localization within these compartments enables US28, US27, and UL33 to bud into a viral membrane when the virus is exiting the cell [49]. Immunohistochemistry on cryosections showed that UL33 is located at multivesicular bodies (MVB), but it is still unknown whether UL33 can recycle back to the PM or it enters a degradation pathway through lysosomes [49].

When co-expressing US27 and US28, colocalization was observed, suggesting that both proteins are located in late endosomes/lysosomes. Based on this overlap, it was examined whether US27 is similarly internalized and revealed that a large amount of this receptor is found in intracellular vesicles, indicating receptor internalization in a constitutive manner, like for US28 [49].

Little is known about UL78. It is believed, that this receptor is not necessary for viral replication in cell cultures, but its function in vivo remains to be studied [49]. This receptor is mainly localized within the cytoplasm of the endoplasmatic reticulum, although surface expression has also been proposed. Furthermore, it has been shown that UL78 undergoes constitutive endocytosis and recycling back to PM [50].

### 3.2. Kaposi’s Sarcoma Associated Herpesvirus (KSHV)

The ORF74 receptor family is encoded by γ2-herpesviruses, such as: KSHV, MHV68, HVS, AtHV, and EHV2. They bind CXC chemokines, and most often also display constitutive activity. ORF74 encoded by KSHV is the most thoroughly characterized receptor in this group, followed by ECRF3 from HVS. Both bind CXC-chemokines; KSHV-ORF74- with a broader spectrum compared to ECRF3-HVS [51,52,53,54,55]. The constitutive signalling of KSHV-ORF74-has been directly linked to tumorigenesis in several mouse models [56,57,58]. In terms of cellular localization, KSHV-ORF74- has been described to be mostly located at the PM (Figure 2), but also to undergo internalization.

CXCL1-, and CXCL8-mediated endocytosis was shown to depend on β-arrestin 1 and 2, whereas the constitutive endocytosis occurs independently of β-arrestin [26]. The AP-2 complex plays an important role in constitutive internalization of ORF74 (Figure 2). This adaptor complex is also important for clathrin-coated vesicles mediated endocytosis [29]. After internalization, the receptor is found in early endosomes, and from there it recycles back at the PM or it fuses with lysosomes and enters the degradation pathway [59].

### 3.3. Epstein-Barr Virus (EBV) and Its Closely Related Lymphocryptoviruses

The first BILF1 receptors were recognized as GPCR’s in 2005 [60]. They are encoded by different γ-1-herpesviruses (EBV, other primate lymphocryptoviruses, porcine lymphotropic herpesviruses (PLHV1, 2, 3) and other ungulate gammaherpesviruses) [61,62,63]. The sequence identity of BILF1 receptors varies, with ape γ-1 herpesvirus BILFs being highly conserved and ungulate γ-1 herpesvirus BILFs being more distantly related [63]. However, ORF encoding BILF1 receptors was described for all above mentioned γ-1 herpesviruses at a similar genomic position [64,65,66]. Functionally, EBV-BILF1 remains the most studied among BILF1 receptors until this day. It is an orphan receptor with well-established constitutive signalling and internalization properties. Confocal microscopy has shown that EBV-BILF1, similar to KSHV-ORF74-, predominantly locates at the cell surface, but there is a difference among BILF1 orthologs of this family [60,63]. Studies of BILF1 receptors from different primate lymphocryptoviruses showed that BILF1 receptor encoded in PtroLCV1 (lymphocryptovirus from chimpanzee) and PpygLCV1 (Lymphocryptovirus from orangutan) predominantly locates intracellularly and not at the PM. It was therefore shown that localization is not preserved among the family [63]. However, EBV-BILF1 has been shown to internalize in a constitutive manner [63], a cellular phenomenon that seems linked to the main function of BILF1 as a MHC-1 downregulating molecule [15]. This immunevasive property has also been confirmed for RhLCV-BILF1, but not for CalHV3-BILF1 [15,67]. Further investigation on BILF1 mediated endocytic routes is therefore needed in order to fully understand its functional properties.

## 4. Methodological Approaches and Novel Techniques to Study Receptor Mediated Endocytosis

GPCR internalization studies are carried out using different methods. The majority of these use the receptor linked with specific epitopes (HA, FLAG-tag, SNAP-tag, CLIP-tag), which enables binding of specific antibodies and therefore receptor detection in cell [68].

### 4.1. Real-Time Internalization Assay

To study the dynamics of internalization and recycling in real-time, a novel method based on time-resolved fluorescence resonance energy transfer (TR-FRET) is used (Figure 3). This method uses an N-terminal receptor tag (called SNAP-tag), a 20 kDa mutant of the DNA repair protein O6-alkylguanine-DNA alkyl transferase. It has no restrictions regarding cellular localization and expression in different cell lines. SNAP-tag is labelled with a tag-lite^®^ SNAP Lumi4^®^-Tb substrate (donor), a cell impermeable substrate chemically inert towards other proteins. After one-hour labelling, the donor is washed off and fluorescein (acceptor) is added to the cells. During the measurement, excitation of the donor causes the energy transfer to acceptor resulting in quenching of the donor emission [69].

With this method, agonist-induced and constitutive internalization can be followed in real-time (Figure 3 and Figure 4). To obtain the best detection of constitutive internalization, labelling of surface receptors with tag-lite^®^ SNAP Lumi4^®^-Tb must be performed at 4 °C to prevent endocytosis during the labelling process. Further, to allow for receptor internalization; measurements are performed at 37 °C. When observing agonist-induced internalization, labelling is usually performed at 37 °C, since the internalization is later triggered by addition of the agonist [69]. This novel sensitive method offers cost effective, fast, high throughput approaches to study ligand-induced or constitutive receptor internalization kinetics. Its limitation is the use of relatively large SNAP-tag on receptors, which could potentially change receptor localization and signalling patterns. Therefore, an additional comparison of SNAP-tagged and WT receptor, as for example receptor signalling, is needed prior the assay.

As an example of this technique applied for the vGPCRs, we SNAP-tagged EBV-BILF1 and the three porcine herpesvirus-encoded (PLHV1-3) BILF1 receptors and determined constitutive internalization of all four receptors (Figure 4). For receptors, labelled with the donor molecule at 37 °C, only a small increase in internalization was observed. This is mainly because the labelled receptors had already reached an equilibrium of the internalization/recycling process during the labelling period. Therefore, we performed the labelling of receptor at 4 °C to prevent any constitutive internalization in this period. With this approach, the internalization was increased.

### 4.2. Antibody Feeding Assay

Antibody feeding allows determination of receptor endocytosis at different time points on fixated cells. In principle, studied receptors must be specifically labelled with antibodies that recognize specific epitopes on the N-terminal site of the receptors. Use of recombinant receptors tagged with engineered extracellular tags (for example FLAG, HA, Myc) are very common and specific antibodies against these tags are commercially available.

Antibody feeding can be performed using two different principles. With ELISA assays, one of the most useful methods for determination of receptor surface expression, the amount of surface expressed receptors can be quantitatively assessed over time. Secondly, using a microscopy approach, surface expression, and internalized receptors, they can be visualized in different time points after induction of internalization. Internalization can be induced by ligands targeting the studied receptors or in case of constitutively active receptors by inducing a temperature shift. Thereby, it is possible to prevent internalization by cooling the cells (incubation at 4 °C) and induce internalization by incubating the cells at temperatures of 18 °C or higher (usually 37 °C).

In both principles, initial incubation with specific primary antibodies at 4 °C prevents internalization processes and allows labelling of only surface expressed receptors. Afterwards, cells are incubated at 37 °C (with or without the ligand) for various time points, allowing the receptors to internalize. Following internalization, cells are fixed and in case of ELISA stained with horseradish peroxidase-conjugated secondary antibodies. In the presence of horseradish peroxidase substrate, secondary antibody gives a reaction of the supernatant and its optical density (OD) can be measured with ELISA plate readers. By this we can determine the amount of the receptors expressed at the cell surface in different time points quantitatively.

Performing microscopy experiments, incubation at 37 °C and additional fixation is followed by labelling with the first secondary antibody labelled with a specific fluorophore (for example Alexa 488) that visualizes labelled receptors at the cell surface. Cells are then additionally permeabilized and incubated with another secondary antibody (for example Alexa 647 antibodies) that labels internalized receptors and unlabelled receptors at the cell surface. This way, we can differentiate between the amount of surface remaining receptors and internalized receptors [68,69,70]. Antibody feeding assays provide direct visualisation of receptor internalization at specific time points. Despite these benefits, the procedures require a few washing steps and temperature shifts, which can affect cell morphology and viability.

Antibody feeding has been used to determine constitutive internalization of EBV-BILF1, PtroLCV1-BILF1, PpygLCV1-BILF1, and SsynLCV1-BILF1 where all described receptors localized into intracellular compartments after 30 min [63]. Further, for US28, both internalization and recycling were confirmed by antibody feeding assay using microscopy approach. In the case of recycling, after incubation with the first antibody, cells were washed and additionally re-incubated with another antibody for 1h at 37 °C. Receptors that recycled to the cell surface were stained with the secondary antibody and internalized again [71].

### 4.3. Fluorescence-Activate Cell Sorting (FACS)

FACS can quantitatively determine the decrease of receptors expressed at the cell surface after ligand addition or in case of constitutive internalization by a temperature shift. Receptors can be detected in several different ways. Similar to ELISA, receptors can be labelled by N-terminal epitope-tags and subsequently incubated with specific primary antibodies directly conjugated with a fluorescent dye. Receptors can also be tagged with a fluorescent tag (for example GFP). Changes in surface expression of the receptor can be measured in different time points. With this method, receptor internalization and recycling rates can be estimated accurately, as it allows the measurement of properties on a single cell in solution [72,73].

### 4.4. Microscopy Based Approaches

Currently, fluorescence microscopy is important and represents a constantly evolving tool in the field of bioscience. When studying receptor trafficking, an initial determination of receptor cellular localization is needed. To determine GPCR localization, fluorescence microscopy is usually used. Receptors can be either stained with a fluorescent antibody against a specific tag or tagged with intrinsically fluorescent protein (green fluorescent protein (GFP)). With confocal microscopy, we can collect serial optical sections from thick specimens at around 300 nm resolution, which enables us to determine receptor localization in the cell with good precision. To observe internalization, antibody feeding approaches can be used (as discussed above).

One of the leading approaches in the field of single-molecule fluorescence microscopy is TIRF (Total Internal Reflection Fluorescence). TIRF is especially useful for visualization of molecules at the cell surface, making it ideal for studying GPCRs. It requires the use of fluorophore tags (eGFP, SEP or SNAP) that might disrupt general properties of wild type receptors, therefore functional analysis of tagged receptors is crucial [74,75,76].

For confocal microscopy, more than only one type of antibody can be used in a single assay, enabling us to determine co-localization of different receptors and/or proteins [72]. As a marker for recycling endosomes, antibodies against the transferrin receptor are used; for endosomes/lysosomes, antibodies against LAMP1 or CD63 are used as exemplified by studies on US28, US27, and UL33. US28 co-localized with markers for recycling endosomes and late endosomes/lysosomes, whereas US27 and UL33 significantly co-localized with markers for late endosomes/lysosomes but not with markers for recycling endosomes [49,71]. The Golgi network is marked with antibodies against γ subunit or adaptor complex 1 [49]. For colocalization with the endoplasmic reticulum (ER), the marker calnexin is commonly used. Rab proteins are specific markers of the endocytic route. Different proteins mark different intracellular organelles which play roles in endocytosis. Rab5 is a marker for early endosomes and is a very important protein in the first step of endocytosis. Rab4 and Rab11 are linked to the recycling process, whereas Rab4 also mediates fast and direct recycling to the PM whilst Rab11 mediates slow recycling from recycling endosomes, usually located in the perinuclear region. Rab7 is a marker for late endosomes/lysosomes. It marks the degradation pathway [77].

### 4.5. Manipulation of Endocytic Pathways

#### 4.5.1. Chemical Inhibitors

To better understand different types of endocytosis, researchers have developed different methods to block these processes. By using these pathway inhibitors, it is possible to assess which cell structures are responsible for certain type of endocytosis. Different pharmacological agents can represent good tools for cell manipulation with their direct inhibition of specific processes. Disadvantages are non-specificity and their ability to interfere with other cell processes [78]. CME is blocked by decreased cytosolic pH, which restrain clathrin vesicles on the PM and disables their removal from the PM [68].

Monodansylcadaverine (MDC) is a tissue transglutaminase inhibitor that participates in clathrin assembly and internalization [68]. Hypertonic sucrose is the most widely used agent, which prevents CME, causing cell shrinkage. It is believed that sucrose also affects endocytosis by caveolae, therefore it is a non-specific endocytic blocker. With potassium depletion, clathrin aggregation occurs in the cytoplasm, which prevents its accumulation on PMs. Chlorpromazine mediates the accumulation of adaptor proteins and clathrin on endosomal membranes, thus preventing the assembly of these proteins on PMs [79]. It also prevents the receptor recycling back to the PM in different endocytic pathways [78]. Pitstop2 is a novel transferrin inhibitor, which interrupts binding between terminal domain of clathrin and amphiphysin [68]. Temperature is another factor influencing endocytosis. CME is blocked under 16 °C for example [78].

Regarding the fact that caveolae mediated endocytosis occurs on lipid rafts, cholesterol depletion in the PM blocks caveolae formation [68]. Cyclodextrins are used for this purpose, but besides their influence on endocytic routes, they also change the membrane structure, therefore influencing different endocytic pathways. As an alternative, filipin can be used. By binding to cell surface cholesterol, it prevents the biding of other molecules, but is cytotoxic in higher quantities [78]. Dynamin is important in both clathrin-mediated and caveolae-mediated endocytosis. At the membrane, it pinches-off newly formed vesicles and allows them to release from the surface. Dynasore has been recently described as a chemical inhibitor of dynamin and can be used for manipulation of both clathrin and caveolin-mediated endocytosis [80].

#### 4.5.2. Genetic Manipulation

To avoid nonspecific effects of chemical inhibitors, different genetic approaches have been developed for endocytic inhibition. These approaches alter the expression of proteins, important for normal receptor endocytosis and includes knockout models, overexpression of dominant negative mutants and protein silencing with siRNA.

The best way to determine a role of a specific protein is gene inactivation. There are plenty of different knockout mouse models available, lacking expression of different proteins [81]. For the purpose of endocytic studies, mouse embryonic fibroblasts (MEF) from β-arrestin knockout mice (βarr 1/2-KO cells) are widely used, to determine whether endocytosis is β-arrestin dependent as exemplified for US28 [82].

CRISPR-cas9 (clustered regulatory interspaced palindromic repeats) is a novel method used for gene knockout. This mechanism is a defence mechanism against viruses and other invaders used by prokaryotes. Its simplicity was exploited for use in mammalian cells for the purpose of genome editing. Two components are involved in this mechanism: Cas9 protein, an enzyme which cuts DNA and sgRNA, which determines the site of cutting. SgRNA needs to be complementary to the cutting site [83]. CRISPR/Cas9 system was used to prepare dynamin-1 and β-arrestin knockout cells. They used these cells to evaluate if the absence of these proteins influences endocytosis. Dynamin-1 is an important component, participating in vesicle scission. In its absence clathrin- and caveolin-mediated endocytosis is disabled. For this purpose, 20 bp dynamin sgRNA was created and cloned into vector along with Cas9n. Transfection was used to introduce the vector to the cell. Dynamin-1 expression was later checked with western blot [84].

Downregulation of protein expression and therefore inhibition of specific types of endocytosis can be performed using dominant negative mutants and siRNA constructs. Dominant negative mutants of dynamin (Dyn1 K44A) [85], β-arrestin (β-arrestin 319–418) [86], caveolin (Cav1 S80E) [87], eps15 (Eps15 Δ95/295) [88], and epsin (Epsin 1ΔUIM) [89] are widely used in different studies determining the endocytic pathway. Cav1 S80E, β-arrestin 319–418 and Dyn K44A dominant mutants have previously been used to determine endocytic properties of other GPCRs (e.g., NK1R) [90]. Overexpression of these mutants in cells masks the wild type protein function, but its function is never completely silenced. Also, it was proposed that overexpression of dominant negative mutants may cause other indirect effects in the cell, leading to misinterpreted results [78].

For silencing protein expression, siRNA constructs against different endocytic players are used. To inhibit the clathrin-mediated pathway, siRNA construct against µ2 subunit of AP2 protein was used in US28 stably transfected Hela cells [25]. Silencing the expression of this protein clearly indicated the requirement for µ2-adaptin subunit and therefore CME in US28. However, the time taken to inhibit the expression of these proteins is long (3–7 days). During this time cells can adjust to protein deficiency or can also change the expression of other genes [78].

Morpholino oligomers/morpholino antisense oligonucleotides (MO) have also been used to understand knockdown effects of GPCRs. They are generally the most widely used anti-sense knockdown tool in zebrafish and are also commonly used in other models such as chick embryos and frogs [91]. Inhibition of gene expression is undertaken via blocking translation or splice blocking, preventing assembly of the spliceosome. Although this technology is relatively inexpensive, differing knockdown yields and off-target effects are always a consideration. The use of standard control MO, knockdown rescue experiments and use of several different knockdown MOs can help decipher potential off-target reactions and even increase knockdown levels as differing MOs are used; however, these are time consuming [92]. GPCRs and regulators ranging from gpr161 through to β-arrestin have been knocked down using this technology [93,94].

Naturally, there are also increasing amounts of data pertaining to genetic mutation/variation from published experiments in cell lines through to patients. Whilst the original intention of this work may not directly look at GPCRs, a range of bioinformatics tools means that the data can be analysed in relation to several lines of enquiry. Data from gene expression studies, biological responses, clinical findings, and outcomes data can all be combined, and networks and biomarkers may well be observed from these studies in relation to GPCRs or factors associated with them. Therefore, data mining and the large range of genomics/bioinformatics tools now available provide an invaluable resource.

## 5. Conclusions

GPCRs play important roles in virus life cycle from the primary infection, replication, and latency establishment to various pathological outcomes. In this respect, understanding molecular and cellular behavior is of huge importance. This review has focused on three thoroughly studied vGPCRs (HCMV-US28, KSHV-ORF74, and EBV-BILF1 and homologs of these) in terms of endocytic properties as well as novel methods used for characterization of endocytic properties. Despite their distant genetic relationships, US28, ORF74, and BILF1 receptors have been linked to the development of cancer. ORF74 was described as the driver of KSHV-related malignancies inducing angiogenesis, cellular transformation, and inflammation [56,58,95]. HCMV-US28 has been detected in glioblastoma and colorectal cancers and is described as an onco-modulator constitutively upregulating angiogenesis and proliferation [96,97]. For BILF1, constitutive Gα_i_ dependent signaling has been linked to EBV-mediated cell transformation ability both in vitro and in vivo [98]. The involvement of all three receptor classes in tumor development makes them attractive drug targets for the treatment of virus-related cancers. As shown for US28, one potential strategy to target infected cells is the use of fusion toxin protein (FTP) to specifically target and kill HCMV infected cells [46,47,48]. For US28, a predominantly intracellular localization pattern was described as a consequence of a rapid constitutive endocytosis [49,99]. This ability was used for the delivery of the toxin intracellularly. For US28, β-arrestin independent, constitutive CME has been shown. This is additionally supported by observations that US28 expresses a YXXΦ motif known to interact with AP-2, a protein important for clathrin coat assembly. A similar motif was observed in BILF1 receptors from EBV and PLHV1-3, as well as KSHV-ORF74. BILF1 has recently been described as an immunevasive protein playing a role in downregulating surface expressed MHC class I molecules [15]. With its cell surface localization, BILF1 has been proposed to bind MHC class I molecules and internalize in complex leading to its degradation in lysosomes. BILF1 receptors are a subject of current research interest and further studies into their endocytic properties are currently ongoing and essential in order to further our knowledge.

Given the close link between signaling and internalization, and for both of these phenomenon a close link to the pathophysiology of the viruses, future studies of the internalization properties of these receptors are needed to finally establish how, when, and where to target them from a therapeutic point of view.

## Figures and Tables

**Figure 1 molecules-25-05710-f001:**
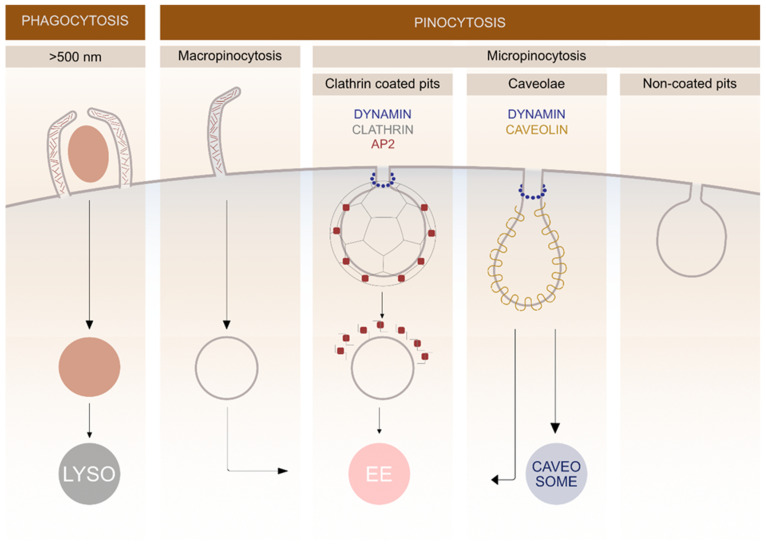
Schematic representation of different endocytic pathways in mammalian cells. The endocytosis is divided into various subgroups based on the size of the cargo entering the cell. Different membrane proteins are involved in clathrin and caveolin-mediated pathways and the fate of cargo molecules depends on specific endocytic mechanisms. Lysosomes (LYSO), adaptor protein 2 (AP2), early endosomes (EE).

**Figure 2 molecules-25-05710-f002:**
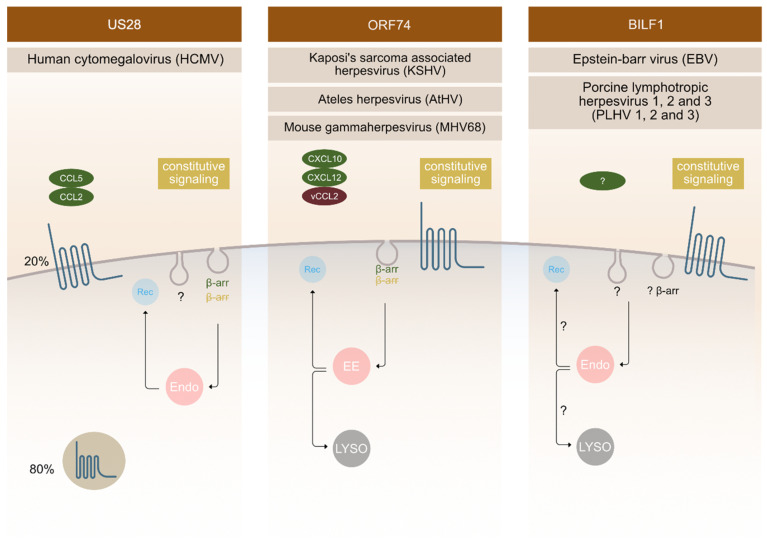
Endocytic mechanisms employed by vGPCRs. Different vGPCRs use different mechanisms to enter the cell. Besides ligand dependent endocytosis (as shown on the Figure for US28 with ligands CCL5 and CCL2 and for ORF74 with ligands CXCL10, CXCL 12 and VCCL2), constitutive (ligand independent) endocytosis is a common feature observed for these receptors. The fate of receptors inside the cell is tightly regulated and has an important impact on receptor function outcome. Localization of vGPCRs differs, with ORF74 and BILF1 receptors predominantly localizing at the surface and US28 localizing intracellularly in 80% and at the surface at 20%. Endosomes (endo), recycling (Rec), β-arrestin (β-arr), early endosomes (EE), lysosomes (LYSO).

**Figure 3 molecules-25-05710-f003:**
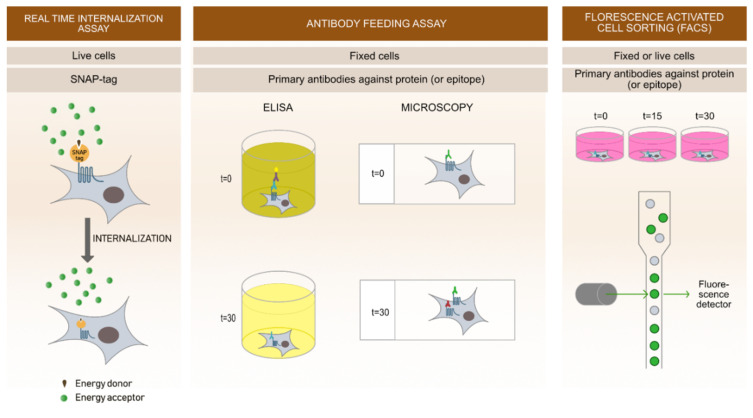
Methods to study GPCR internalization. Different approaches can be used for receptor internalization analysis.

**Figure 4 molecules-25-05710-f004:**
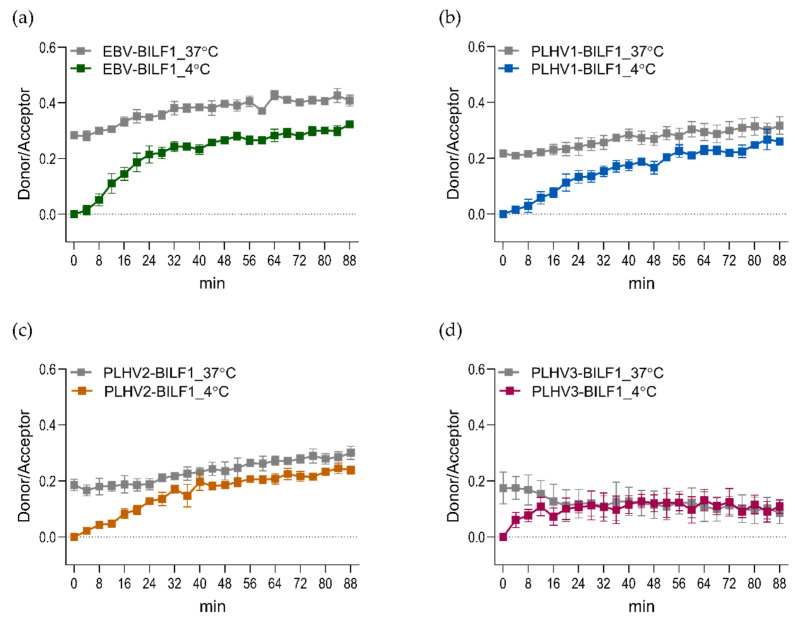
BILF1 receptors internalize constitutively. HEK293A cells were transfected with (**a**) SNAP-EBV-BILF1, (**b**) PLHV1-BILF1, (**c**) PLHV2-BILF1, and (**d**) PLHV3-BILF1. Donor labelling was performed at 4 °C (coloured curves) and 37 °C (grey curves). Graphs represent means ± SEM of two individual experiments performed in triplicate.

**Table 1 molecules-25-05710-t001:** Examples of viral GPCRs (vGPCRs) from different herpesvirus families (β and γ) (Adapted from [16]).

Family	Virus	Receptor	Preferred Endocytic Pathway	Signaling Pathways	G Protein Coupling
β-Herpesviruses	Human cytomegalovirus (CMV)	US27	-	-	-
		US28	β-arrestin independent clathrin-mediated, partly through lipid rafts	Constitutive NFκB, NFAT, CREB, PLC, SRF, STAT3, TCF/LEF, Ligand induced PLC, MAPK	Gα_q_, Gα_i/o,_ Gα_12/13_
		UL33	-	Constitutive SRC, CREB	Gα_q_, Gα_i,_ Gα_s_
		UL78	-	-	-
	Human herpesvirus 6	U12	-	-	-
		U51	-	-	-
	Human herpesvirus 7	U12	-	-	-
		U51	-	-	-
	Mouse cytomegalovirus	M33	-	Constitutive PLC, NFκB, CREB	Gα_s_
		M78	-	-	-
	Rat cytomegalovirus	R33	-	Constitutive PLC, NFκB	Gα_q_, Gα_i_
		R78	-	-	-
γ-Herpesviruses	Human herpesvirus 8 (HHV8 or Kaposi’s sarcoma virus (KSHV))	ORF74	β-arrestin independent clathrin mediated constitutive endocytosis, β-arrestin dependent clathrin-mediated ligand dependent endocytosis	Constitutive and ligand induced RAC1, PLC, PKC, AKT, JNK-SAPK, LYN-SRC, GSK3, JAK2-STAT3, HIF1α, PI3Kγ, calcineurin	Gα_q_, Gα_i_,Gα_12/13_
	Ateles herpesvirus (AtHV)	ORF74-AtHV	-	-	-
	MouseHV68	ORF74-MHV68	-	not constitutively active PLC, MAPK, Akt, NFκB	Gα_i_
	Equine HV2 (EHV2)	E1	-	-	-
		E6	-	-	-
		ORF74-EHV2	-	-	Gα_i_
	Herpesvirus Saimiri (HVS)	ECRF3	-	-	-
	Human Epstein barr virus (EBV)	BILF1	-	Constitutive NFκB, NFAT, CREB	Gα_i_
	Rhesus lymphocryptovirus (RhLCV)	BILF1	-	Constitutive NFκB	Gα_i_
	Callitrichine herpesvirus 3 (CalHv3)	BILF1	-	Constitutive NFκB, NFAT	Gα_i_
	Pan troglodytes lymphocryptovirus 1 (PtroLCV1)	BILF1	-	Constitutive NFκB	Gα_i_
	Gorilla gorilla lymphocryptovirus 1 (GgorLCV1)	BILF1	-	-	-
	Gorilla gorilla lymphocryptovirus 2 (GgorLCV2)	BILF1	-	-	-
	Pongo pygmaeus lymphocryptovirus 1 (PpygLCV1)	BILF1	-	Constitutive NFκB	Gα_i_
	Pongo pygmaeus lymphocryptovirus 2 (PpygLCV2)	BILF1	-	-	-
	Symphalangus syndactylus lymphocryptovirus 1 (SsynLCV1)	BILF1	-	Constitutive NFκB, NFAT	Gα_i_
	Symphalangus syndactylus lymphocryptovirus 2 (SsynLCV2)	BILF1	-	-	-
	Macaca fascicularis lymphocryptovirus 1 (MfasLCV1)	BILF1	-	-	-
	Erythrocebus patas lymphocryptovirus 1 (EpatLCV1)	BILF1	-	-	-
	Piliocolobus badius lymphocryptovirus 1 (PbadLCV1)	BILF1	-	-	-
	Ateles paniscus lymphocryptovirus 1 (ApanLCV1)	BILF1	-	-	-
	Pithecia pithecia lymphocryptovirus 1 (Ppit LCV1)	BILF1	-	-	-
	Porcine lymphotropic herpesvirus 1, 2 and 3 (PLHV1-3)	BILF1	-	-	-

protein kinase B (Akt), cAMP responsive element binding protein (CREB), glycogen synthase kinase 3 (GSK3), hypoxia inducible factor 1α (HIF1α), Janus kinase 2 (JAK2), c-jun N-terminal kinase (JNK), lymphocyte enhancing factor (LEF), tyrosine-protein kinase (LYN), mitogen-activated protein kinase (MAPK), nuclear factor of activated T-cells (NFAT), Nuclear factor kappa B (NF-κB), phosphatoinositide-3-kinase-γ polypeptide (PI3Kγ), protein kinase C (PKC), phospholipase C (PLC), Ras-related C3 botulinum toxin substrate 1 (RAC1), stress-activated protein kinase (SAPK), serum response factor (SRF), signal transducer and activator of transcription 3 (STAT3), T-cell factor (TCF).

**Table 2 molecules-25-05710-t002:** Predicted eukaryotic linear motifs (ELM) in vGPCRs.

Receptor	Elm Name	Instances	Positions	Elm Description	Cell Compartment
		(Matched Sequence)			
**EBV-BILF1**	TRG_ENDOCYTIC_2	YSAF	32–35 [A]	Tyrosine-based sorting signal responsible for the interaction with µ2 subunit of AP (Adaptor Protein) complex	plasma membrane,
					clathrin-coated endocytic vesicle,
					cytosol
**PLHV1-BILF1**	TRG_ENDOCYTIC_2	YTTL	179–182 [A]	Tyrosine-based sorting signal responsible for the interaction with µ2 subunit of AP (Adaptor Protein) complex	plasma membrane,
					clathrin-coated endocytic vesicle,
					cytosol
**PLHV2-BILF1**	TRG_ENDOCYTIC_2	YAVL	159–162 [A]	Tyrosine-based sorting signal responsible for the interaction with µ2 subunit of AP (Adaptor Protein) complex	plasma membrane,
					clathrin-coated endocytic vesicle,
					cytosol
**PLHV3-BILF1**	TRG_ENDOCYTIC_2	YAAL	194–197 [A]	Tyrosine-based sorting signal responsible for the interaction with µ2 subunit of AP (Adaptor Protein) complex	plasma membrane,
					clathrin-coated endocytic vesicle,
					cytosol
**US28**	TRG_ENDOCYTIC_2	YYAI	130–133 [A]	Tyrosine-based sorting signal responsible for the interaction with µ2 subunit of AP (Adaptor Protein) complex	plasma membrane,
		YAIV	131–134 [A]		clathrin-coated endocytic vesicle,
		YRPV	138–141 [A]		cytosol
		YDYL	177–180 [A]		
		YLEV	179–182 [A]		
		YHSM	321–324 [A]		
**ORF74**	TRG_ENDOCYTIC_2	YGLF	326–329 [A]	Tyrosine-based sorting signal responsible for the interaction with µ2 subunit of AP (Adaptor Protein) complex	plasma membrane,
					clathrin-coated endocytic vesicle,
					cytosol

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
