# Peer review of "Methods for Studying Endocytotic Pathways of Herpesvirus Encoded G Protein-Coupled Receptors"

_molecules, 2020, doi:10.3390/molecules25235710_

Round 1

Reviewer 1 Report

The review by Cho et al. focuses on the endocytic pathways of virus encoded G protein-coupled receptors (vGPCRs).  It also covers contemporary methods used in examining GPCR internalization. The subject matter is of interest to those who study viral entry and signal transduction. The coverage on endocytosis and its relevancy to vGPCR function is comprehensive. However, the two major parts are somewhat discordant.  The authors may wish to provide specific examples to illustrate how the different methods described were used in the field of vGPCRs. Other suggestions are given below.

Major Comments

  • Although the authors provided an overview of the different vGPCRs and various endocytic pathways that are known to date, there is little information on which vGPCR prefers which pathway. It would be useful to have such information included in Table 1. Also, the authors should briefly describe the difference, if any, among the large number of BILF1 receptor types shown in Table 1. Are there functional or structural differences?
  • For the layman, it might be helpful to indicate whether the preferred endocytic pathway of a vGPCR has any functional implication on viral infection or replication. Given the similarity of vGPCRs with host GPCRs, what are the predominant signaling pathways for each of the receptors listed in Table 1? What are their G protein coupling specificities? Such information can be included in Table 1.
  • When describing the different endocytic pathways, the authors should relate them to the vGPCRs. Examples of how a particular technique enabled/facilitated the study of vGPCR internalization will greatly improve the manuscript.
  • There are quite a few typo/grammatical errors that need to be corrected (please see Minor Comments).

Minor Comments

  • Abstract should be in one single paragraph.
  • Line 38 should read as: by which a cell…
  • Line 53 should read as: a mechanism used by several…………for initial entry…
  • Citation of Table 1 should be at the end of the preceding paragraph, not as a new paragraph.
  • Please reformat Table 1 to reduce the number of horizontal lines.
  • Line 94 should read as: After endosome maturation,
  • Line 97 should read as: occurs, providing newly…
  • Line 135 should read as: GTPases are…
  • Line 190: delete “GPCR”
  • Line 230 should read as: shown to depend…
  • Line 231 should read as: endocytosis occurs independently of…
  • Line 241: delete “By”
  • Line 230 should read as: shown to depend…
  • Figure 2: some examples of ligands shown in the figure (e.g., CXCL10 and CXCL12) are not mentioned in the text. Should align with what the text.
  • Table 2: entries in the last two columns are essentially identical and could be removed and simply described in the text.
  • Section 5.1 reads like a general description for section 5. It is just a short paragraph introducing the subsequent sections. Perhaps it is better to delete the section heading and move all subsequent sections up by one.
  • Line 277: Figure 4 (some labels are far too small) should be named as Figure 3 by sequence of appearance.
  • Line 340 should read as: Similar to ELISA, receptors…
  • Line 342 meaning not clear for the sentence “Receptors can……with fluorescent dyes.”
  • Line 379-380 should read as: Disadvantages are…
  • Line 423 should read as: inhibition of specific…
  • Line 436 should read as: During this time…
  • Line 459 should read as: has focused…

Reviewer 2 Report

This is a well written up-to-date review that covers the enocytic pathways for herpes virus-encoded GPRCs.

Minor comments:

  1. Please use the term 'clathrin-coated vesicle' rather than clathrin vesicle. Lines 111, 130, 193, 233
  2. Please change 'clathrin mediated endocytosis' to 'clathrin-mediated endocytosis' on lines 110 and 114.
  3. Lines 129-130 should read 'with ubiquitin-interacting motifs'. Please remove on their surface as these are short motifs.
  4. In the sentence on lines 142-43 it is not clear which cargo motifs the vGPCRs carry that are recognised by clathrin adaptor proteins. A clarification would be helpful. This could be facilitated by moving the paragraph on lines 258-267 up (including Table 2). Please include a reference to Table 2 in the main body of the text.
  5. Please use the Greek letter for phi in the text when discussing the YxxΦ motif. Please define which amino acids that Φ represents (line 127).
